# Targeting HR Repair as a Synthetic Lethal Approach to Increase DNA Damage Sensitivity by a RAD52 Inhibitor in BRCA2-Deficient Cancer Cells

**DOI:** 10.3390/ijms22094422

**Published:** 2021-04-23

**Authors:** Wei-Che Tseng, Chi-Yuan Chen, Ching-Yuh Chern, Chu-An Wang, Wen-Chih Lee, Ying-Chih Chi, Shu-Fang Cheng, Yi-Tsen Kuo, Ya-Chen Chiu, Shih-Ting Tseng, Pei-Ya Lin, Shou-Jhen Liou, Yi-Chen Li, Chin-Chuan Chen

**Affiliations:** 1Graduate Institute of Natural Products, Chang Gung University, Taoyuan 333, Taiwan; dean19901207@hotmail.com (W.-C.T.); wazai1023@gmail.com (S.-F.C.); kikilala827@hotmail.com (Y.-T.K.); yachen502@gmail.com (Y.-C.C.); megting064@gmail.com (S.-T.T.); ruby700512@gmail.com (P.-Y.L.); gibson@ms8.url.com.tw (S.-J.L.); 2Tissue Bank, Chang Gung Memorial Hospital, Taoyuan 333, Taiwan; d49417002@gmail.com; 3Graduate Institute of Health Industry Technology and Research Center for Chinese Herbal Medicine, College of Human Ecology, Chang Gung University of Science and Technology, Taoyuan 333, Taiwan; 4Department of Applied Chemistry, National Chiayi University, Chiayi 600, Taiwan; cychern@mail.ncyu.edu.tw (C.-Y.C.); whaha26@gmail.com (Y.-C.L.); 5Department of Physiology, College of Medicine, National Cheng Kung University, Tainan 701, Taiwan; violet69320@gmail.com; 6Translational Research Program in Pediatric Orthopedics, The Children’s Hospital of Philadelphia, Philadelphia, PA 19104, USA; mosquito1213@gmail.com; 7Cryo-EM Center, Vagelos College of Physicians and Surgeons, Columbia University Irving Medical Center, New York, NY 10032, USA; yc3642@cumc.columbia.edu

**Keywords:** DNA repair, curcumin, RAD52 inhibitor, synthetic lethal, BRCA-deficient

## Abstract

BRCA mutation, one of the most common types of mutations in breast and ovarian cancer, has been suggested to be synthetically lethal with depletion of RAD52. Pharmacologically inhibiting RAD52 specifically eradicates BRCA-deficient cancer cells. In this study, we demonstrated that curcumin, a plant polyphenol, sensitizes BRCA2-deficient cells to CPT-11 by impairing RAD52 recombinase in MCF7 cells. More specifically, in MCF7-siBRCA2 cells, curcumin reduced homologous recombination, resulting in tumor growth suppression. Furthermore, a BRCA2-deficient cell line, Capan1, became resistant to CPT-11 when BRCA2 was reintroduced. In vivo, xenograft model studies showed that curcumin combined with CPT-11 reduced the growth of BRCA2-knockout MCF7 tumors but not MCF7 tumors. In conclusion, our data indicate that curcumin, which has RAD52 inhibitor activity, is a promising candidate for sensitizing BRCA2-deficient cells to DNA damage-based cancer therapies.

## 1. Introduction

Double-strand breaks (DSBs) are the most severe type of DNA damage. Chromosome rearrangement and cell death are thought to be consequences of a failure to accurately repair DSBs [1]. The repair of DSBs is mediated via the homologous recombination (HR) or nonhomologous end joining (NHEJ) pathway. NHEJ occurs mainly in the G1 phase of the cell cycle, while HR is the major DNA repair mechanism occurring in the S and G2/M phases. Since most normal cells stop dividing in the G1 phase and cancer cells are more often in the S phase, the HR pathway is more frequently used to repair DNA damage [2]. HR-mediated repair begins with the recognition and binding of DSB ends by the MRN complex, which consists of the MRE11, RAD50, and NBS1 proteins. The DSB is then resected by MRE11 and the 5′-3′ exonuclease EXO1 to generate single-stranded DNA via a process called DNA end resection. Subsequent binding of the RPA protein to single-stranded DNA (RPA-ssDNA) attracts additional repair proteins such as BRCA1 and BRCA2. BRCA1 and BRCA2 help load RAD51 onto the RPA–ssDNA complex to identify homologous sequences of sister chromatids and initiate HR repair [3,4,5].

Chemotherapy and radiation therapy have been reported to induce a large number of DSBs. Activation of cell death in cancer cells due to unrepaired damage suggests that it is possible to improve the efficacy of these therapies by enhancing the sensitivity of cancer cells to DSBs [6,7]. Combining chemotherapy or radiation therapy with targeted inhibitors of the DNA repair pathway was shown to increase treatment efficacy [8,9]. Since most cancers have different defects in their response to DNA damage [10], the use of targeted inhibitors of the DNA repair pathway is a specific method to target only cancer cells without affecting healthy cells [11]. PARP inhibitors (PARPis) are the most extensively used anticancer agents that suppress the growth of BRCA-deficient cancer cells and increase their sensitivity to chemotherapy drugs [7,12]. Indeed, a clinical study has shown that triple-negative breast cancer (TNBC), especially carrying BRCA mutations, has a higher response to PARP inhibitors. This result indicated that targeting HR repair is a promising means to sensitize BRCA-mutated TCNB cells [13]. The development of PARPis has resulted in revitalized research interest regarding other constituents of the HR pathway. In terms of the HR pathway, RAD52 has been proposed as another target to sensitize BRCA-deficient cancer cells.

RAD52 is known to play a major role in DNA repair in yeast, and yeast cells deficient in RAD52 are unable to repair DNA damage, resulting in cell death [14]. However, deletion of RAD52 does not produce a phenotype in human cells or other mammalian cells [15]. New findings that RAD52 is essential for cell viability in BRCA1-, PALB2-, BRCA2- and RAD51 paralog-deficient cells but not in normal cells have suggested that RAD52 may represent an attractive therapeutic target for killing breast cancer and ovarian cancer cells [11]. Cells with either BRCA2 deficiency or RAD52 deletion survived, suggesting that Rad52 function may overlap with that of BRCA2, while simultaneous deletion of both genes was lethal [16]. These data suggested that in addition to PARP inhibitors, RAD52 inhibitors may be another choice for the treatment of cancer in the future [15,17].

There have been numerous clinical trials with successful outcomes in which PARPi showed synthetic lethality with “BRCAness” cancer [18]. Hence, the concept of synthetic lethality induced by pharmaceuticals is a promising approach for personalized therapy. However, drug resistance to PARPi may occur. Thus, novel strategies/components to treat cancer are urgently needed to kill cancer cells rapidly before mechanisms of resistance develop. Recently, RAD52 has emerged as a precision therapy for BRCA-deficient cells. In BRCA2-deficient cells, RAD52 plays a backup role that stabilizes the RAD51 filament. Targeting RAD52 could be a potential personalized therapy for BRCA2-deficient cancer [19]. Furthermore, a novel strategy called “dual synthetic lethality” has been proposed to target two different repair pathways [15]. Studies have shown that simultaneously targeting RAD52 while treating patients with PARPis specifically eradicates BRCA-deficient cells without damaging agents [11]. This indicates that the development of RAD52i could be used for the treatment of BRCA-deficient cancer cells or as a dual synthetically lethal agent.

Curcumin is one of the main components of turmeric (Curcuma longa) and has been shown to have a synergistic effect with chemotherapeutic drugs such as irinotecan (CPT-11) or radiation therapy [20,21]. However, the mechanism remains unclear. CPT-11, a cytotoxic plant alkaloid, has been reported to inhibit DNA topoisomerase I and induce DSBs in cells [22,23]. CPT-11 has been clinically used as an anticancer drug for the treatment of colorectal cancer, cervical cancer, ovarian cancer, and breast cancer [24]. A number of groups, including ours, previously showed that curcumin increased the sensitivity of cancer cells to DNA damage by inhibiting the DNA damage response (DDR) [25,26]. We were also the first to report that curcumin increased the sensitivity of yeast to DSBs by suppressing the expression of RAD52 protein [27].

Our study showed that curcumin, a RAD52 inhibitor, effectively inhibited the HR repair pathway and the proliferation of BRCA2-deficient cells. Curcumin also increased the sensitivity of BRCA2-deficient cells to DNA-damaging drugs. In contrast, curcumin failed to inhibit the growth of BRCA2-overexpressing cells, thereby increasing the resistance of BRCA2-overexpressing cells to DNA-damaging drugs. Furthermore, curcumin reduced tumor growth in BRCA2-deficient cells following CPT-11 treatment. Our results suggested that curcumin suppressed the growth of BRCA2-deficient cells by inhibiting the expression of RAD52 and increasing the sensitivity of BRCA2-deficient cells to DNA-damaging drugs.

## 2. Results

### 2.1. Curcumin Impaired the Expression and Foci Formation of RAD52 Induced by CPT-11 in MCF7 Cells

We have previously shown that curcumin increases DNA damage sensitivity by inhibiting the expression of RAD52 in budding yeast [27]. Here, we first analyzed whether curcumin inhibited RAD52 in the breast tumor cell line MCF7 following DNA-damaging agent treatment. MCF7 cells were treated with 0.2 μM CPT-11 with or without various concentrations of curcumin. On the basis of immunoblot analysis, we found curcumin to impair the CPT-11-induced upregulation of RAD52 expression in a dose- and time-dependent manner, whereas the expression of MRE11 was not affected by curcumin (Figure 1A,B). Furthermore, we determined the effect of curcumin on the ability of MCF7 cells to form nuclear repair foci in response to CPT-11 by immunofluorescence with antibodies against RAD52, an important enzyme in HR [17]. RAD52-containing repair foci were visible in MCF7 cells following CPT-11 treatment. However, the combination of CPT-11 and curcumin evoked significant decreases in RAD52 foci formation (Figure 1C). Hence, curcumin decreased the expression and foci formation of RAD52 in response to CPT-11.

### 2.2. Curcumin Inhibited the Expression of RAD52 Recombinase Following CPT-11 Treatment

We next investigated whether curcumin affects the transcript level of RAD52 following CPT-11 treatment. The mRNA levels of RAD52 were analyzed by RT-PCR and RT-QPCR. As shown in Figure 2A,B, the induced mRNA levels of RAD52 were downregulated with 40 µM curcumin treatment in MCF7 cells. Thus, curcumin treatment decreased the transcript levels of RAD52 in response to CPT-11. In our previous study, we demonstrated that curcumin could impair RAD52 protein expression through proteasome-mediated proteolysis [27]. To further investigate whether proteasome-mediated degradation of RAD52 is elicited by curcumin in MCF7 cells, MG132, a 26 S proteasome inhibitor, was added to MCF7 cells treated with both CPT-11 and curcumin. We found that MG132 treatment attenuated the decrease in RAD52 in cells cotreated with CPT-11 and curcumin (Figure 2C). To further evaluate the posttranslational decreases in RAD52, we conducted molecular docking analysis. The docking results were displayed by Discover Studio, and the crystal structure of the RAD52 protein (PDB id: 1kn0) was docked with curcumin. The docking analysis showed docking potential with the N-terminus of RAD52. On the basis of the bonding type, bonding distance, and position of the curcumin structure, we selected possible results (Figure 2D). Curcumin showed hydrogen bonds that crossed the different monomers of PHE26 in the ring structure of RAD52, which could lead to a conformational change in RAD52 (Figure 2E). These results indicated that as a RAD52 inhibitor, curcumin transcriptionally and post-translationally decreases RAD52 expression in response to CPT-11.

### 2.3. Curcumin Specifically Sensitized BRCA2-Deficient Cells to CPT-11

Rad52 is known to play a major role in DSB repair in yeast. Yeast cells deficient in Rad52 are unable to repair DSBs, resulting in cell death [14]. Although deletion of RAD52 does not produce a phenotype in human cells or other mammalian cells [15], previous studies have suggested that Rad52 inactivation in BRCA2-deficient cells results in synthetic lethality [16]; this suggests that RAD52 is a potential target for the treatment of BRCA2-deficient tumors. Therefore, we proposed that the inhibition of Rad52 by curcumin could show synthetic lethality in BRCA2-deficient cells; furthermore, we hypothesized that curcumin could specifically sensitize BRCA2-deficient cells to CPT-11. To determine whether BRCA2-deficient cells are more sensitive to curcumin than BRCA2-wild-type cells, we performed clonogenic survival assays on MCF7-siControl and MCF7-siBRAC2 cells (Figure 3A,B). After transfection, siBRCA2-2 significantly decreased the expression of BRCA2; therefore, siBRAC2-2 was used for BRCA2 knockdown in further experiments (Appendix A). Without CPT-11 treatment, MCF7-siBRAC2 cells were more sensitive to curcumin treatment than MCF7-siControl cells (Figure 3C). Moreover, the inhibition of Rad52 by curcumin specifically sensitized MCF7-siBRAC2 cells but not BRCA2-wild-type cells to CPT-11, indicating that curcumin significantly sensitized MCF7-siBRAC2 cells to CPT-11 (Figure 3D,E).

### 2.4. Curcumin Impaired the Repair System of BRCA2-Deficient Cells, Resulting in Genomic Instability

To further investigate whether curcumin sensitizes MCF7-siBRAC2 cells to CPT-11 by impairing the repair of damaged DNA, we performed a single-cell gel electrophoresis assay (or comet assay) to detect DNA damage persisting after recovery from CPT-11 treatment. MCF7-siBRAC2 cells had more unrepaired DSBs than MCF7-siControl cells, as indicated by the longer comet tail in MCF7-siBRAC2 cells following curcumin treatment (Figure 4A). The relative extent of unrepaired DSBs was quantified by calculating the average tail moment, which is a measure of the amount of DNA present in the comet tail (Figure 4B). These results suggest that curcumin contributed to the inhibition of DSB repair, and as a consequence, MCF7-siBRAC2 cells were more sensitive to CPT-11 than MCF7-siControl cells following curcumin treatment.

### 2.5. Curcumin Inhibited HR to Sensitize BRCA2-Deficient Cells

To maintain genome integrity, cells initiate repair systems in response to DNA damage. In HR repair, BRCA2/RAD52 plays an important role in loading RAD51 onto ssDNA, facilitating strain invasion. When the cells have defects in both BRCA2 and RAD52, their repair system is impaired, which generates unrepaired damage in the cell, eventually leading to cell cycle arrest. RAD52 has been reported to have a key role in HR when BRCA2 is inactive [16]. To determine whether the inhibition of RAD52 by curcumin can dramatically impair HR in BRCA2-deficient cells, we utilized the pDR-GFP recombination reporter assay. In brief, a DSB is introduced into a truncated GFP gene by expressing SceI endonuclease, and repair of the break by HR restores the expression of the GFP gene, which can be measured by flow cytometry (Figure 5A). The expression of BRCA2 protein level was assayed by immunoblotting (Figure 5B). MCF7-shBRAC2 cells demonstrated a lower HR rate (2.64%) than MCF7-shControl cells (3.51%) following curcumin treatment, which reduced RAD52 expression (Figure 5C,D). These data suggest that curcumin dramatically impaired HR in BRCA2-deficient cells.

### 2.6. Curcumin Decreased the Sensitivity to CPT-11 in Capan1 Cells That Overexpress Exogenous BRCA2

Next, we further confirmed that the inhibition of RAD52 by curcumin sensitized BRCA2-deficient cells to CPT-11. We proposed that overexpression of BRCA2 may counteract the sensitization of BRCA-deficient cancer cells to CPT-11-induced DNA damage after curcumin treatment. To explore this question, we employed pancreatic adenocarcinoma Capan1 cells with mutations in the BRCA2 gene. To restore BRCA2 function, we transfected Capan1 cells with BRCA2 or empty vector. Transfection with BRCA2 resulted in a significant increase in BRCA2 protein expression in Capan1 cells (Figure 5E). Through clonogenic survival assays, we found that curcumin increased the sensitivity of Capan1 cells to CPT-11 (Figure 5F), which is consistent with previous results that curcumin sensitizes BRCA2-deficient cancer cells to DNA damaging drugs (Figure 3). Furthermore, the sensitization of Capan1 cells to curcumin was counteracted in Capan1-BRCA2 cells (Figure 5G). These data strengthen our hypothesis that curcumin selectively sensitizes BRCA2-deficient cells to chemotherapy.

### 2.7. Curcumin Sensitized BRCA2-Knockout MCF7 Cells to Chemotherapy In Vivo

To further investigate the cellular and molecular findings that curcumin sensitizes BRCA2-deficient cells described above, we examined the therapeutic efficacy in terms of tumor size in tumor xenograft mice. Thus, the flanks of nude mice were subcutaneously injected with 2 × 10^6^ MCF7 or BRCA2-knockout (BRCA2 KO) MCF7 cells, while tumors reached approximately 100 mm^3^. Mice were intraperitoneally injected with vehicle, curcumin (55 mg/kg), CPT-11 (10 mg/kg), or a combination of curcumin and CPT-11 (Figure 6A). Capan1 is a BRCA2-deficient pancreatic cancer cell line as a negative control. The protein level of BRCA2 was verified by Western blot. (Figure 6B). Furthermore, the BRCA2 KO cell line did not exhibit changes in cell proliferation (Appendix A). As shown in Figure 6C,F, the body weight revealed that none of the dosages and treatments in this study had pernicious effects on mice. Tumor growth was significantly decreased following curcumin, CPT-11, and combined treatment compared to the vehicle control in the MCF7 xenograft model at day 16. However, the growth of tumors did not show a difference between CPT-11 and combined treatment (Figure 6D,E). Furthermore, we observed the same effect on the BRCA2 KO xenograft model in which curcumin and CPT-11 reduced tumor growth at day 16. Interestingly, combining CPT-11 with curcumin markedly reduced tumor growth, indicating that curcumin sensitized BRAC2-deficient cells to CPT-11 treatment (Figure 6G,H). To determine the in vivo effect of curcumin on RAD52 expression, we subjected tumor sections to immunohistochemistry (IHC). RAD52 expression was decreased following curcumin treatment, and IHC data showed that the expression of RAD52 following curcumin treatment was lower than that following vehicle or CPT-11 treatment alone (Figure 6I), which is consistent with an in vitro study showing that curcumin decreased the RAD52 expression level (Figure 1). In conclusion, curcumin, as a RAD52 inhibitor, was able to sensitize BRCA2-deficient cells to CPT-11.

## 3. Discussion

We noticed that curcumin per se did not affect MCF siCcontrol cells in the clonogenic assay at low concentrations (the IC_50_ of curcumin is 54.7 uM, Appendix A); however, curcumin per se still reduced tumor growth in mice. Previous studies showed that curcumin inhibits tumor formation among the dosages from 40 to 300 mg/kg in mice [28,29,30]. We were not surprised that curcumin per se inhibits tumor load at the dosage of 55mg/kg according to references. Although we may have used a lower dosage, we intended to emphasize that a relatively low dosage of curcumin and CPT-11 could additively impair tumor load in mice. Indeed, the data showed that combining CPT-11 with curcumin markedly reduced tumor growth (Figure 6).

Previous studies have shown that inactivating RAD52 predominantly suppresses the growth of BRCA-deficient cells [16]. We demonstrated that curcumin increased DNA damage sensitivity by impairing RAD52 in BRCA2-deficient cells (Figure 3A,B). Moreover, exogenous BRCA2 expression in Capan1 cells reversed the sensitivity to CPT-11-induced DNA damage with curcumin treatment (Figure 5G). These data support our idea that curcumin exerts a growth inhibitory effect against BRCA2-deficient cells by impairing RAD52. In this study, the dosage that we used only partially inactivated RAD52 in Capan1 cells (Appendix A), and BRCA2 knockdown was incomplete in MCF7-siBRCA2 cells. The effect of synthetic lethality may not have been as eminent as in previous studies. However, these data did show the promising characteristic that curcumin is a novel RAD52 inhibitor that can be used as an alternative treatment. Furthermore, we used RNA interference to knockdown BRCA2 expression to increase DNA damage sensitivity in response to curcumin treatment. However, the BRCA2 functional defect in cancer cells could be truncation, mutation, or secondary mutation [3,31]. Therefore, the type of BRCA dysfunction that could lead to DNA damage sensitivity by curcumin needs to be further evaluated.

Our data demonstrated that RAD52 expression decreased at the transcriptional and posttranslational levels following curcumin treatment (Figure 2). In previous studies, it has been shown that curcumin inhibits the activity of the GATA-2, GATA-3, and ZIC2 transcription factors, which in turn decreases the transcription level of RAD52 [32,33,34,35]. Furthermore, the data shown in Figure 2E confirm that curcumin docked with the N-terminus of RAD52, possibly causing a conformational change that blocks the dsDNA and ssDNA binding properties of RAD52. Furthermore, the N-terminus of RAD52 is involved in sumoylation triggered by the MRE11–Rad50–Mrx2 complex and N-terminal simulation, allowing Rad52 to protect against proteasomal degradation [36,37]. Curcumin may block the sumoylation of RAD52, presuming that RAD52 could lead to proteasomal degradation. To conclude, the docking results and previous studies might explain why curcumin decreased RAD52 expression at the posttranslational and transcriptional levels in our study (Figure 2C).

The heptameric ring of RAD52 has been extensively studied, and each domain has been elucidated recently. The RAD52 monomer, in order to exert its function, is released to the cytoplasm and the region of the N-terminal response for oligomerization to form the ring structure, being responsible for DNA/RNA binding [38,39]. In contrast, the C-terminal domain encompasses the area that interacts with replication protein A (RPA), RAD51 recombinase is responsible for HR, and the nuclear localization signal (NLS) yields RAD52 oligomer transport to the nucleus [15,40,41,42]. Previous studies have shown that cells deficient in BRCA rely on RAD52 for robust DNA repair and that RAD52 is synthetically lethal with BRCA1, BRCA2, RAD51 paralogs, and PALB2 [14,16]. Hence, RAD52 has emerged as a promising target for personalized anticancer treatment [15]. Seven kinds of RAD52i have been proposed, and some of these compounds are approved by the FDA [43], indicating that RAD52i is an attractive approach for BRCA1-, BRCA2-, RAD51 paralog-, and PALB2-deficient cancer [3,14]. A previous study confirmed that BRCA1/2-deficient leukemia cells were inhibited by the RAD52 inhibitor F79, which interferes with the DNA binding of RAD52 [44]. F79 treatment showed a synergistic effect with imatinib in BRCA1/2-deficient breast, ovarian, and pancreatic cancer. The compound 6-OH-dopa disrupts RAD52 heptameter formation, leading to inhibition of Rad52 recruitment to damage sites [45]. Huang et al. identified that D-I03 reduces the level of SSA repair and suppresses the proliferation of BRCA-deficient cells following cisplatin treatment [46]. Compound F779-0434 was designed with a high affinity for RAD52 and interacts with residue Lys152 on RAD52, which plays a pivotal role in ssDNA binding [47]. Furthermore, the FDA-approved RAD52 inhibitor A5MP can hamper ssDNA binding and halt BRCA1-deficient cell proliferation [43]. Another compound, ZMP, which is known to emulate the function of A5 MP, disrupts RAD52-ssDNA foci in BRCA1-deficient cells [15]. RAD52, which is redundant with BRCA2, specifically targets BRCA-deficient cells with limited effects on normal tissues. Thus, RAD52i is a promising therapeutic approach for BRCA-deficient cancer. In this study, the docking data showed that the binding of curcumin was required to link the different dimers of RAD52, indicating that curcumin only disrupts the heptameric ring of RAD52. Curcumin docked to the N-terminus of RAD52 could impair ssDNA/RNA binding, which is one of the main functions of RAD52. In addition, compared to other RAD52i cells, curcumin has malignant cell growth inhibitory characteristics, is nontoxic to normal tissues, and has anti-inflammatory properties.

In conclusion, curcumin—a main ingredient in turmeric that has been incorporated into the human diet for thousands of years—has recently been proposed to have anticancer properties. However, the RAD52 inhibition property of curcumin has not yet been elucidated, except with our previous findings in yeast. In this study, we revealed that curcumin is a potential RAD52 inhibitor that increases DNA damage sensitivity in BRCA2-deficient cancer.

## 4. Materials and Methods

### 4.1. Drugs, Chemicals, and Plasmids

Curcumin (≥94% curcuminoid content) and irinotecan (CPT-11) were purchased from Sigma-Aldrich. MG132 was obtained from Santa Cruz. All of these drugs were dissolved in dimethyl sulfoxide (DMSO). pcDNA3 236HSC WT (BRCA2) was a gift from Mien-Chie Hung (Addgene plasmid # 16246; http://n2t.net/addgene:16246; RRID: Addgene_16246). pDRGFP was a gift from Maria Jasin (Addgene plasmid # 26475; http://n2t.net/addgene:26475; RRID: Addgene_26475). pcBASceI was a gift from Maria Jasin (Addgene plasmid # 26477; http://n2t.net/addgene:26477; RRID: Addgene_26477).

### 4.2. Cell Culture and Reagents

MCF7 and Capan-1 cells were provided by Dr. Shu-Hei Wang and Dr. Pei-Hsin Huang, respectively (National Taiwan University). Cells were cultured in Dulbecco’s modified Eagle’s medium (Invitrogen, CA, USA) supplemented with 10% fetal bovine serum (Gibco, NY, USA) and 1% penicillin/streptomycin (Invitrogen) at 37 °C with 5% CO_2_. Cell lines used in this study have been tested as mycoplasma-free and authenticated.

### 4.3. BRCA2-Knockdown Cell Line Generation

To generate the BRCA2-knockout MCF7 cell line, we conducted CRISPR/Cas9 to target the BRCA2 gene (Santa Cruz Biotechnology sc-400700, CA, USA). The transfected cells were seeded onto 96-well plates and selected with puromycin (4 mg/mL). Full-length BRCA2 was detected in puromycin-selected clones by Western blotting, and the *BRCA2 KO* MCF7 cell line was generated.

### 4.4. BRCA2 Transfection

For transient depletion of BRCA2 in MCF7 cells, we purchased siRNA against BRCA2 (sc-29825) and control (sc-37007) from Santa Cruz Biotech (CA, USA). For stable knockdown, we transfected MCF7 cells with plasmid constitutively expressing shRNA, which was selected with puromycin for at least 2 weeks. All transfections were performed with Lipofectamine 2000 (Invitrogen, CA, USA). pLKO-shBRCA2 (TRCN0000009825) was obtained from the RNAi Core Facility, Academia Sinica (Taipei, Taiwan). pLKO-shLacZ, pLKO-shLuc, and pc-DNA were gifts from Dr. Szu-Hua Pan (National Taiwan University), and pCIN BRCA2 WT was a gift from Dr. Mien-Chie Hung (Addgene plasmid #16245, USA).

### 4.5. Homologous Recombination Reporter Assay

After treatment with drugs for 4 h, MCF7-shRNA cells were co-transfected with pDR-GFP and pCBASceI for 2 days using Lipofectamine (Invitrogen, CA, USA). The percentage of GFP-positive cells was determined by flow cytometry on a BD FACSC Alibur flow cytometer, and FlowJo software was used for analysis. pDRGFP was a gift from Maria Jasin (Addgene plasmid # 26475; http://n2t.net/addgene:26475; RRID:Addgene_26475) andpCBASceI was a gift from Maria Jasin (Addgene plasmid # 26477; http://n2t.net/addgene:26477; RRID:Addgene_26477). All values are moralized to cells that transfected with control shRNA or shBRCA2 without curcumin treatment.

### 4.6. Clonogenic Assay

MCF7 cells were transfected with siRNA for 24 h and then seeded in 6-well plates in triplicate at the desired cell density (500 to 1000 cells) for 16 h. The cells were then treated with CPT-11 to induce DNA breaks and combined with curcumin for 1 hour. Next, CPT-11 was removed, and cells were cultured in media containing curcumin for 10–14 days at 37  °C to allow colonies to form. Colonies were stained with 0.5% crystal violet/25% methanol and counted. To define the colony size and number, we used the OpenCFU software to calculate colonies.

### 4.7. Immunoblot Analysis

Cell extracts were processed in Laemmli buffer (125 mM Tris-HCl (pH 6.8), 20% glycerol, 4% SDS). The samples were resolved by electrophoresis on a 10% SDS-polyacrylamide gel. The following primary antibodies were used for immunoblotting at a 1:1000 dilution: anti-RAD52 (5H9) was purchased from GeneTex, anti-BRCA2 (234403, CA, USA) was purchased from R&D Systems, anti-Mre11 (C-16) was purchased from Santa Cruz Biotechnology, anti-β-Actin (AC-15, CA, USA) was purchased from Sigma-Aldrich, and anti-mouse (A9044, MO, USA) and anti-goat (A5420, MO, USA) HRP-linked secondary antibodies were purchased from Sigma-Aldrich and used at a 1:10,000 dilution. Images were acquired with a Wealtac KETA-CL.

### 4.8. Immunofluorescence

In brief, MCF7 cells were seeded onto poly-L-lysine-coated coverslips at a density of 3 × 105 cells and incubated overnight. After treatment with drugs for 12 h, the cells were fixed with 4% formaldehyde (*w*/*v*) in PBS for 15 min. Coverslips were washed with PBS and then immunostained with primary antibodies against RAD52 (Cell Signaling #3425). Appropriate Alexa Fluor-594 (red)-conjugated secondary antibody (1:200) was purchased from Thermo Scientific. Slides were viewed with a Nikon ECLIPSE Ni-U plus fluorescence microscope.

### 4.9. Comet Assay

MCF7-siRNA cells were treated with CPT-11 for 1 h before being collected for the assay. Cell gel electrophoresis under alkaline conditions was performed for approximately 20 min at 25 V. Samples stained with DAPI (Abcam, ab104139, UK) were observed using a Nikon ECLIPSE Ni-U plus fluorescence microscope. Images were analyzed using Open Comet software.

### 4.10. Apoptosis Assay

For annexin V/PI assays, cells were stained with annexin V–FITC and PI and evaluated for apoptosis by flow cytometry according to the manufacturer’s protocol (Molecular Probes apoptosis assay kit,Thermo Fisher, V13241, MA, USA). Briefly, 5 × 10^5^ cells were washed once with PBS and stained with 2 µL annexin V–FITC and 5 µL of PI (5 mg/mL) in 1 × binding buffer (10 mM HEPES (pH 7.4), 140 mM NaCl, 2.5 mM CaCl_2_) for 20 min at room temperature in the dark. Apoptotic cells were determined using a BD FACSCalibur. Both early apoptotic (annexin V-positive, PI-negative) and late (annexin V-positive, PI-positive) apoptotic cells were included in cell death determinations.

### 4.11. Xenograft Tumor

The experimental protocol was approved by the Institutional Animal Care and Use Committee of Chang Gung University (IACUC protocol no. CGU108-138). Nude mice (5 weeks of age) were provided by the National Laboratory Animal Center (NLAC), NAR Labs, Taiwan. Mice were held for 7 days for acclimation before the experiment was conducted and subcutaneously implanted with 2 × 106 MCF7 or BRCA2-knockout MCF7 cells in serum-free medium into the dorsal flank region. All mice were housed in individual cages and maintained at room temperature at 23 ± 1 °C with a 12-h dark/light cycle. When the size of the tumor reached approximately 100 mm^3^, the mice were randomized into groups for the following treatment. CPT-11 and curcumin were dissolved in saline containing 10% DMSO, and mice bearing tumors received vehicle intraperitoneally at 10% DMSO, CTP-11 at 10 mg/kg, curcumin at 55 mg/kg, or a combination of CPT-11 and curcumin every 2 days. Tumor size was measured and calculated every 2 days on the basis of the following formula: tumor volume (length width2)/2. After 16 days, the mice were euthanized with CO_2_ inhalation for 5 min following the secondary (confirmatory) method of cervical dislocation, and xenograft tumors were resected for the following experiment. The tumors were diced and fixed with 4% paraformaldehyde or flash frozen in liquid nitrogen for immunohistochemistry and immunoblotting, respectively. RAD52 immunohistochemistry was performed using FFPE tissue sections of tumors. Slides were first incubated for 1 h at 65 °C and then deparaffinized in xylene, rehydrated in graded ethanol solutions, and finally boiled in Trilogy reagent (Cell Marque, Rocklin, CA, USA) for 10 min for antigen retrieval. The slides were immersed in 3% hydrogen peroxide for 10 min after washing with 1 × PBS. After triple rinsing with 1 × PBS, the sections were exposed to the anti-RAD52 antibody and incubated with slides (1:100) for 1 h at room temperature. Sides were rinsed with 1 × PBS 3 times and then incubated with a biotinylated secondary antibody (Dako, Glostrup, Denmark) for 25 min. After rinsing with 1 × PBS, the slides were treated with horseradish peroxidase-conjugated streptavidin for 25 min at room temperature. The peroxidase activity was developed with 3,30-diaminobenzidine (DAB, Dako), followed by counterstaining with hematoxylin.

### 4.12. Cell Viability Assay

Cell viability assay was performed as previously described [47]. Cell viability assay was carried out by plating 5000 cells per well into 96-well plates. In the following day, cells were treated with various concentrations of curcumin and incubated for 24 h. Cell viability was measured using MTT assay. For MTT assay, 5 μL of MTT (4 mg/mL) solution was added to the cells in each well containing 200 μL of medium. After incubating at 37 °C for 3 h, the supernatant was removed and 100 μL of DMSO was added to the cells. The MTT color reaction was examined using a microplate reader set at A560 nm. 

## Figures and Tables

**Figure 1 ijms-22-04422-f001:**
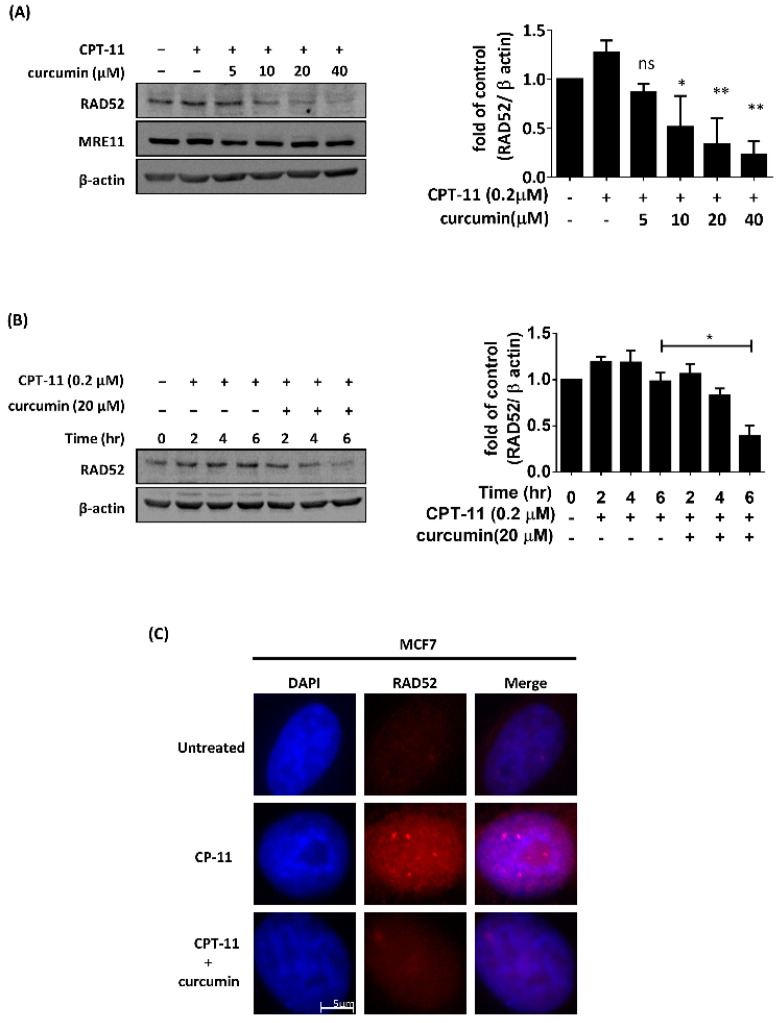
Curcumin inhibited the protein expression and recruitment of RAD52 recombinase following CPT-11 treatment in MCF7 cells. (**A**) Left: MCF7 cells were treated with 0.2 μM CPT-11 combined with different concentrations of curcumin. After 6 h, cells were processed for Western blot analysis using RAD52, MRE11, and β-actin antibodies. Right: quantification of the immunoblotting result. (**B**) Left: MCF7 cells were treated with 0.2 μM CPT-11 combined with or without 20 µM curcumin at different time points. Cells were then processed for Western blot analysis using RAD52 and β-actin antibodies. Right: quantification of the immunoblotting result. Asterisks (*) indicate significant differences between CPT-11-treated and CPT-11 + curcumin-treated cells (* *p* < 0.05; ** *p* < 0.01). (**C**) MCF7 cells were treated with 0.2 μM CPT-11 combined with or without 20 µM curcumin. After 6 h, cells were processed for immunofluorescence analysis using RAD52 antibody to identify foci. Nuclear staining was performed with DAPI.

**Figure 2 ijms-22-04422-f002:**
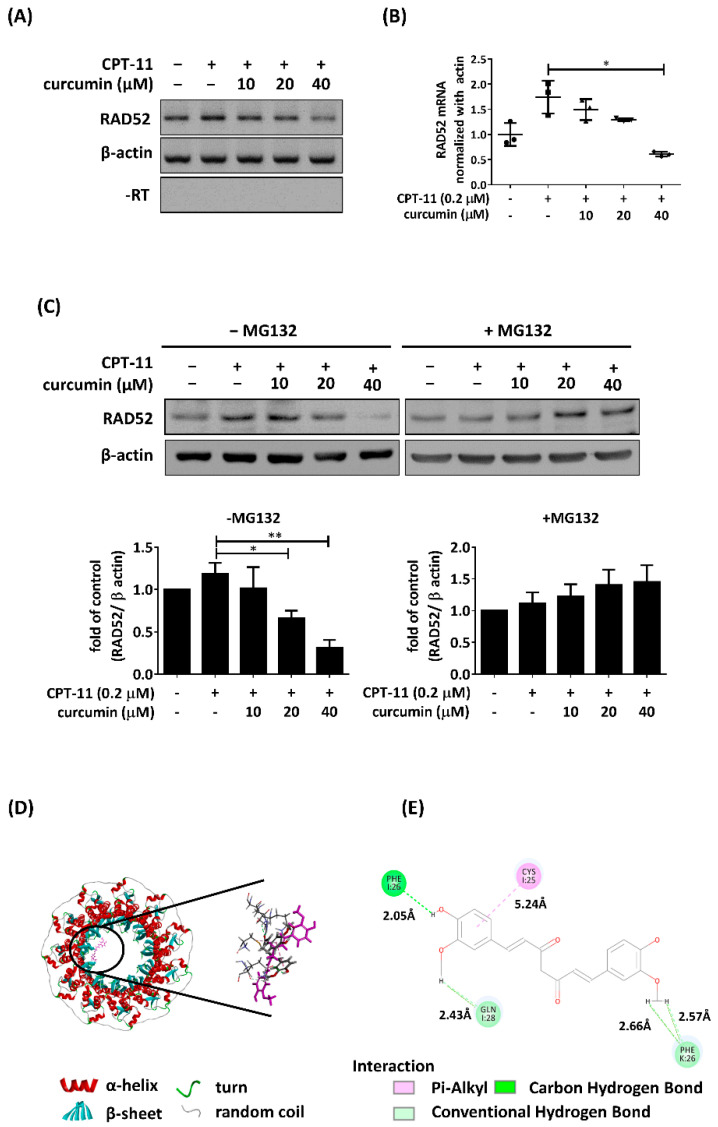
Curcumin translationally and post-translationally regulated RAD52 recombinase following CPT-11 treatment in MCF7 cells. (**A**) MCF7 cells were treated with 0.2 μM CPT-11 combined with different concentrations of curcumin. After 6 h, total RNA was processed for reverse transcription to generate cDNA and analyzed with PCR and (**B**) qPCR. (**C**) MCF7 cells were cotreated with various concentrations of curcumin and 0.2 μM CPT-11 in the presence of MG132. After 6 h, whole-cell extracts were collected for Western blot analysis. Bottom: quantification of the immunoblotting result. (**D**) A docking model of curcumin in the N-terminal domain of RAD52 shown as a ribbon (PDB ID: 1kn0). (**E**) Ligand–protein interactions with the binding residues of RAD52 and curcumin. The green dashed lines indicate hydrogen bonds, and the pink dashed lines indicate π interactions. Two-dimensional diagram showing the conventional hydrogen bond to Phe26 (2.05 Å), the carbon–hydrogen bond to Gln28 (2.43 Å) and to Phe26 (2.66 Å and 2.57 Å), and the π–alkyl interaction to Cys25 (5.24 Å). Asterisks (*) indicate significant differences between CPT-11 and combined CPT-11 and curcumin cells (* *p* < 0.05, ** *p* < 0.01).

**Figure 3 ijms-22-04422-f003:**
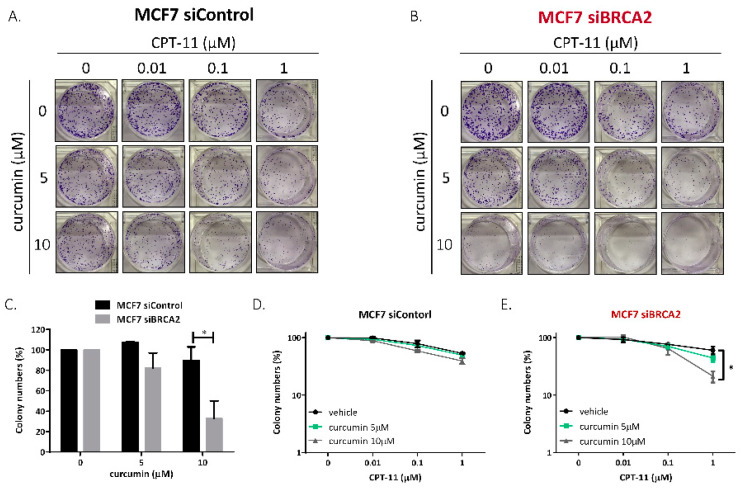
Curcumin specifically sensitized BRCA2-deficient cells to CPT. (**A**,**B**) Clonogenic assay. Transfected cells were grown for 10–14 days before fixation and staining with crystal violet. The detailed procedures are described in the Materials and Methods section. (**C**) MCF7-siBRCA2 cells were more sensitive to curcumin treatment than MCF7-siControl cells. Quantification of colony numbers from the first columns of Figure 3A,B. (**D**,**E**) Curcumin sensitized BRCA2-deficient cells to CPT-11 treatment. Quantification of colony numbers from A and B. Asterisks (*) indicate significant differences between CPT-11 and combined CPT-11 and curcumin cells (* *p* < 0.05).

**Figure 4 ijms-22-04422-f004:**
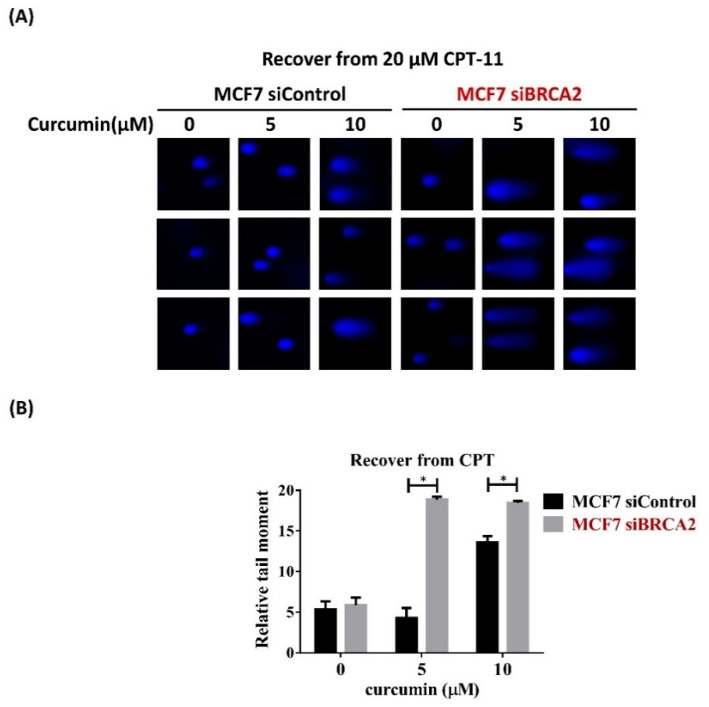
Curcumin reduced HR to sensitize BRCA2-deficient cells to chemotherapy. Curcumin inhibited DNA repair following CPT-11 treatment in MCF7-siBRCA2 cells. (**A**) The cells were treated with CPT-11 to induce DSBs for 1 h; then, CPT-11 was washed out, and the cells were allowed to recover in media containing curcumin for 4 h. The samples were analyzed by comet assay. (**B**) Quantification of the tail moment of over 50 cells from A by open comet. Error bars represent the standard deviations of at least 3 independent experiments. Asterisks (*) indicate significant differences between CPT-11 and combined CPT-11 and curcumin cells (* *p* < 0.05).

**Figure 5 ijms-22-04422-f005:**
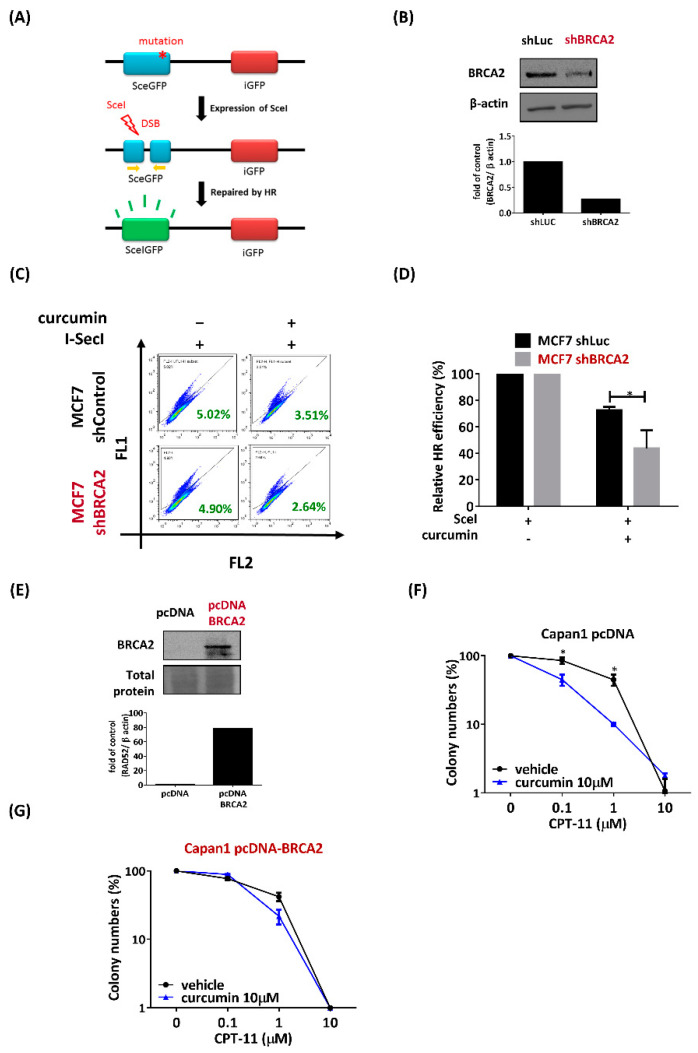
Curcumin decreased the sensitivity to CPT-11 in Capan1 cells that overexpress exogenous BRCA2. (**A**) The scheme illustrates the consequences of I-SceI-induced DSBs in DR-GFP. pSceI-GFP is a truncated GFP coding region that contains an I-SceI cutting site (blue box). The cleavage of the I-SceI cutting site can be repaired by homologous combination with a downstream GFP donor (red box) and results in GFP-positive cells (green box). (**B**) Immunoblot analysis of BRCA2 was performed to confirm knockdown. Bottom: quantification of the immunoblotting result. (**C**) Curcumin decreased HR in the MCF7-shBRCA2 cell line. MCF7-shRNA cells were cotransfected with I-SceI and DR-GFP plus with or without 20 µM curcumin. Forty-eight hours later, the cells were analyzed by flow cytometry to determine the percentage of GFP-positive cells. (**D**) The bar graph shows the mean percentage of GFP-positive cells. All values are moralized to cells that were transfected with control shRNA or shBRCA2 without curcumin treatment. (**E**) BRCA2-deficient Capan1 cells were transfected with pcDNA or pcDNA-BRCA2 plasmid. Cell samples were taken and processed for immunoblot analysis using a BRCA2 antibody. Bottom: quantification of the immunoblotting result. (**F**,**G**) Quantification of colony numbers from the clonogenic assay (data not shown). After transfection with the pcDNA or pcDNA-BRCA2 plasmid, Capan1 cells were seeded in six-well plates and then grown in media containing CPT-11 alone or combined with curcumin for 10-14 days. The detailed procedures are described in the Materials and Methods section. Asterisks (*) indicate significant differences between CPT-11 and combined CPT-11 and curcumin cells (* *p* < 0.05).

**Figure 6 ijms-22-04422-f006:**
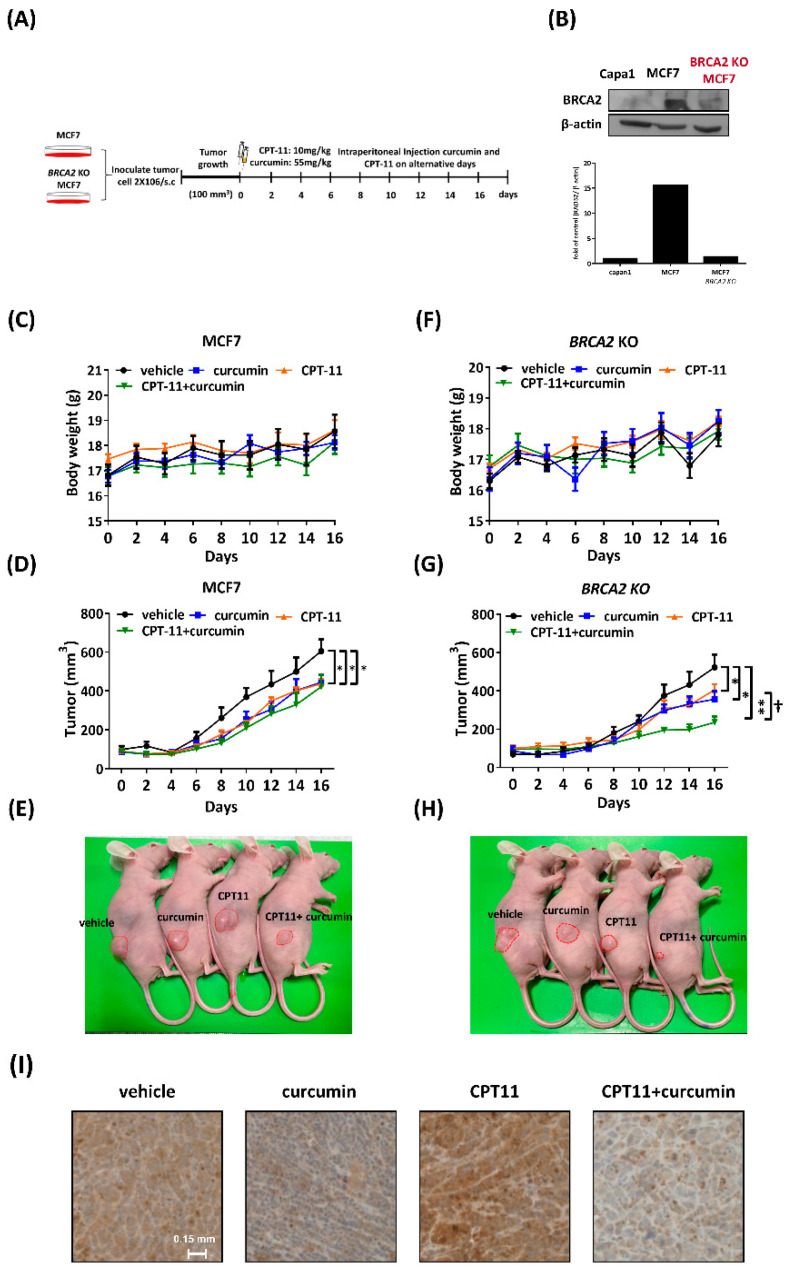
Curcumin sensitized BRCA2 KO MCF7 cells to chemotherapy in vivo. (**A**) Experimental procedure for subcutaneous implantation of MCF7 or BRCA2 KO MCF7 cells into the right flank of nude mice. The mice were then treated with vehicle, CPT-11 (i.p., 10 mg/kg), curcumin (i.p., 55 mg/kg), or their combination every other day for 16 days (*n* = 5). Tumor growth was monitored every 2 days, and the volume was calculated with the formula: volume  =  (width)^2^  ×  length/2. (**B**) BRCA2 expression levels were evaluated by Western blot using anti-BRCA2 and anti-actin antibodies. Bottom: quantification of the immunoblotting result. (**C**) The body weight of the MCF7 xenograft model was recorded every 2 days. (**D**) The tumor growth and volume of the MCF7 xenograft model were recorded every 2 days, and (**E**) representative images of the MCF7 tumor from each group are shown. Asterisks (*) indicate significant differences between the vehicle and other groups (* *p* < 0.05). (**F**) The body weight of the BRCA2 KO MCF7 xenograft model was recorded every 2 days. (**G**) The tumor growth and volume of the BRCA2 KO MCF7 xenograft model were recorded every 2 days. (**H**) Representative images of BRCA2 KO MCF7 tumors from each group. Asterisks (*) indicate significant differences between the vehicle and other groups (* *p* < 0.05, ** *p* < 0.01). (^†^) indicates significant differences between the CPT-11 and combined groups (^†^
*p* < 0.05). (**I**) Immunohistochemistry (IHC) for RAD52 in the xenograft mouse model of BRCA2 KO MCF7 tumors.

## Data Availability

The data presented in this study are available on request from the corresponding author.

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
