# Peer review of "Targeting HR Repair as a Synthetic Lethal Approach to Increase DNA Damage Sensitivity by a RAD52 Inhibitor in BRCA2-Deficient Cancer Cells"

_ijms, 2021, doi:10.3390/ijms22094422_

Round 1
Reviewer 1 Report
The work entitled “Targeting HR repair as synthetic lethal approach to increase 2 DNA damage sensitivity by a RAD52 inhibitor in BRCA2-defection cancer cells” by Tseng et al. reports the RAD52 inhibitory effects of curcumin resulting in increased DNA damage sensitivity of BRCA-2 deficient cancer cells. The work is well planned and illustrated.
I have the following concerns:
- I do not see an abstract of the work. Please include.
- Densitometric analysis for all the western blots shown in the paper should be included.
- In Fig 1B label CPT-11 treatment properly. The arrow looks out of place.
- What is the IC-50 of curcumin for MCF-7 cells at mentioned time points?
- In Fig 2B the CPT-11 dose is 4 µM while the legend mentions as 0.2 µM, please address the discrepancy.
- In Fig 2C which panel of western blot represent MG132 treatment as both are labelled as -MG132? Also, why the beta-actin of first panel is not equal?
- Legend of Fig 2 mentions panel D and E as A and B, please correct it.
- Fig 3 panel D and E represent cell viability? or colony numbers as mentioned in legend? Why the axis of the line graphs starts from 10 (as origin) in these panels?
- In Fig panel F and G, the line graphs starts with 1 as origin. Please follow one format for all the graphs in the figures.
- In Fig 6B the first lane of western blot is labelled as capa1. what is capa1?
- The discussion needs to be trimmed and some of the information may be moved to the introduction.
- Define the colony size in material and methods.
- Just confirming in the title of the paper the BRCA-2 defection (deficient??) is correctly mentioned?
Reviewer 2 Report
The authors have demonstrated that curcumin sensitizes BRCA2 deficient cells to a chemotherapeutic agent- CPT 11 by impairing homologous recombination. The study is nicely conducted and can be accepted if the authors make certain corrections to it.
Comments:
- Figure 2C- The blot panels are not labeled correctly. Both the panel says "-MG132". Considering that the second panel is "+MG132", it is not clear why the addition of MG132 will cause any decrease in RAD52 expression (compare first lanes of the first and second panel of blots). Since MG132 is a proteasome inhibitor, why RAD52 will degrade more upon the addition of MG132. Also, in the first panel, it does not look like any difference in the band's intensity of RAD52. In addition, beta-actin expression is very low in the last lane of the first panel of the blots. The authors should provide better blots with densitometry analysis here.
- Please correct the legends' labeling in figure 2. The legends D and E are labeled as A and B.
- The authors did not observe any significant effects of curcumin on the MCF cell lines in the clonogenic assay; however, in vivo data shows that curcumin was as effective as CPT-11 in reducing tumor load. Please discuss the discrepancies between in vitro and in vivo observations. Although the authors stated that curcumin somewhat inhibited colony formation in MCF cells, it was not significant and not comparable to the effects mediated by CPT-11 treatment. It is hard to believe that curcumin was as effective as CPT-11 in reducing mice's tumor load. Did the authors use a very high concentration of curcumin in the mice that will cause such reduction? Please provide an explanation.
- Page 7, line 198- The authors compare the HR rate of MCF-shcontrol cells to the curcumin-treated MCF-shBRCA2 cells, which is not the right comparison. Instead, the authors should compare the HR rate of curcumin-treated MCF-shControl cells (3.51%) and curcumin-treated MCF-shBRCA2 cells (2.64%).
- It is not clear how authors measure HR rate? Are they determining the percentage of GFP-positive cells? If yes, why is the number so high in figure D? Please explain the difference in figure 5C and D.
Round 2
Reviewer 2 Report
The authors' replies to the comments have been found satisfactory.